# The Gut-Prostate Axis: A New Perspective of Prostate Cancer Biology through the Gut Microbiome

**DOI:** 10.3390/cancers15051375

**Published:** 2023-02-21

**Authors:** Kazutoshi Fujita, Makoto Matsushita, Marco A. De Velasco, Koji Hatano, Takafumi Minami, Norio Nonomura, Hirotsugu Uemura

**Affiliations:** 1Department of Urology, Kindai University Faculty of Medicine, Osakasayama 589-8511, Japan; 2Department of Urology, Osaka University Graduate School of Medicine, Suita 565-0871, Japan; 3Department of Genome Biology, Kindai University Faculty of Medicine, Osakasayama 589-8511, Japan

**Keywords:** gut, microbiome, prostate cancer, castration resistant prostate cancer, microbiota

## Abstract

**Simple Summary:**

The gut microbiome plays important roles in the development of several diseases. The gut microbiome is a dynamic system that is affected by several factors, such as dietary habits, and since prostate cancer and diet are closely linked, it is reasonable to hypothesize that a gut microbiome—affected by diet—could regulate prostate cancer far from the gut, thus creating a gut-prostate axis. Gut dysbiosis result in the leakage of gut bacterial metabolites, such as short-chain fatty acids and lipopolysaccharide into the systemic circulation, leading to the prostate cancer growth. Patients with prostate cancer have a distinct gut microbiome. Furthermore, the gut microbiome produces androgen, affecting castration-resistance of prostate cancer. The gut-prostate axis could be a new target for the prevention and management of human prostate cancer.

**Abstract:**

Obesity and a high-fat diet are risk factors associated with prostate cancer, and lifestyle, especially diet, impacts the gut microbiome. The gut microbiome plays important roles in the development of several diseases, such as Alzheimer’s disease, rheumatoid arthritis, and colon cancer. The analysis of feces from patients with prostate cancer by 16S rRNA sequencing has uncovered various associations between altered gut microbiomes and prostate cancer. Gut dysbiosis caused by the leakage of gut bacterial metabolites, such as short-chain fatty acids and lipopolysaccharide results in prostate cancer growth. Gut microbiota also play a role in the metabolism of androgen which could affect castration-resistant prostate cancer. Moreover, men with high-risk prostate cancer share a specific gut microbiome and treatments such as androgen-deprivation therapy alter the gut microbiome in a manner that favors prostate cancer growth. Thus, implementing interventions aiming to modify lifestyle or altering the gut microbiome with prebiotics or probiotics may curtail the development of prostate cancer. From this perspective, the “Gut–Prostate Axis” plays a fundamental bidirectional role in prostate cancer biology and should be considered when screening and treating prostate cancer patients.

## 1. Introduction

The incidence of advanced prostate cancer has been increasing in the USA and Japan, although the incidence of all prostate cancer in the USA has decreased since the recommendation of the US preventive service task force for prostate-specific antigen (PSA) screening in 2012 [1]. Prostate cancer develops in patients in their 50 s and is a slow-growing tumor. Prostate cancer develops due to the mutations in driver genes, but several factors, such as genetic background and lifestyle affect the development of prostate cancer. Overall, the incidence of prostate cancer is high in Western countries compared with that in Asia. Japanese who lived in Hawaii had a higher incidence of prostate cancer compared with Japanese who lived in Japan [2]. Obesity is also associated with the incidence of prostate cancer. A meta-analysis showed that obese men had a higher risk of advanced prostate cancer [3], and dietary habits are one of the major factors affecting prostate cancer. Western-style diets, especially a high-saturated-fat diet, are associated with increased risk of prostate cancer [4]. High-fat diets (HFD) promote local inflammation in prostate tumors which leads to the increased secretion of inflammatory cytokines, such as IL-6 that culminates into increased recruitment and infiltration of myeloid-derived suppressor cells (MDSCs) [5]. Il-6 is a pleiotropic cytokine that in this context, also promotes prostate cancer growth by activating STAT3 signaling. However, the mechanisms as to how consuming a high-fat diet leads to local tumor inflammation remain largely unknown. For some time, the mechanisms that link dietary habits to prostate cancer remained elusive; however, recent advances in technology have allowed us to come closer to unraveling the mystery. Particularly, developments in next generation sequencing have enabled us to probe deeper. For example, we have been able to capture whole profiles of microbes through 16S rRNA sequencing. By having a better understanding of gut microbiota, we are in a better position to discover associations of the gut microbiome with various diseases. The gut microbiome is a dynamic system that is affected by several factors, such as dietary habits, and since prostate cancer and diet are closely linked, it is reasonable to hypothesized that a gut microbiome—affected by diet—could regulate prostate cancer far from the gut, thus creating a gut-prostate axis. We have just begun connecting the dots that exist between the gut microbiome and prostate cancer, and more and more, additional dots are being discovered and linked. In this review, we discuss the current status of the “gut-prostate axis” as regulated by the gut microbiome.

## 2. The Gut Microbiome

Trillions of microorganisms and their genetic material constitute the gut microbiome. The typical gut contains 10^13^ to 10^14^ bacteria, and these microorganisms are important as they supply nutrients, such as essential amino acids and vitamins, that cannot be produced by humans [6]. Gut bacteria ferment dietary fiber and produce monosaccharides and short-chain fatty acids (SCFAs), such as butyrate, acetate, propionate, and isopropionate [7]. Humans are then able to utilize absorbed acetate and propionate as substrates for lipid, glucose, and cholesterol metabolism [8,9].

Herbivores have a long gastrointestinal tract and harbor gut microbes that allow them to digest cellulose and other plant materials to produce short-chain fatty acids as an energy source, and produce amino acids from ammonia by fermentation in the gut lumen [10]. The gut microbiota also coevolves with the host [11]. For instance, the giant panda consumes bamboo as an energy source, but the gastrointestinal tract of the giant panda is short and simple. Genetic sequencing showed that giant pandas have all the genes necessary for digesting meat, but not cellulose, which is the main component of bamboo [12]. Metagenomic analysis of gut microbes of giant pandas revealed putative genes, encoding cellulose-digesting enzymes and hemicellulose-digesting enzyme from gut microbes indicating that this adaptation enabled them to use bamboo as an energy source [13].

Gut microbiota is affected by several factors. Genetic background is an innate factor that affects the gut microbiome and this was demonstrated with a study analyzing the gut microbiome from twins which showed that host genetics do indeed influence the composition of the human gut microbiome [14]. A genome-wide association study of the host genome revealed 31 loci that affect the microbiome, and the lactase gene locus showed a significant age-dependent association with the abundance of *Bifidobacterium* [15]. In addition to innate factors, perinatal factors contribute to the developing gut microbiome. Everyone’s gut microbiome is established early during the neonatal stage. The neonate gut microbiome is affected by the amniotic fluid, maternal lifestyle, and maternal exposure to antibiotics even before birth. After birth, each individual develops a unique gut microbiome based on their sex, race, and lifestyle. Sex differences affect gut microbial composition in mice and humans. Young adult women in Western countries had a higher diversity of gut microbiota than men, but these differences were not found in Chinese individuals [16]. Testosterone also affects the gut microbiome, and the composition of the gut microbiome changes gradually with puberty. In pubertal subjects, the abundance of the genera *Adlercreutzia*, *Ruminococcus*, *Dorea, Clostridium*, and *Parabacteroides* is associated with the levels of testosterone [17]. Androgen receptor signalling is involved in the maintenance of diversity of gut microbiota; androgen-receptor knock-out mice experienced HFD-induced metabolic syndrome and gut dysbiosis, but intervention of the gut microbiome by antibiotics prevented metabolic syndrome [18].

Lifestyle is one of the main factors affecting gut microbiota. Results from 16S rRNA sequencing of feces has revealed that clusters of individuals with similar lifestyles share similar gut microbiomes. For example, family members share a similar gut microbiome, and obesity is associated with phylum-level changes in gut microbiota [19]. Further evidence of this phenomenon is characterized by differences of the gut microbiome observed between individuals from Western and non-industrialized countries [20]. For instance, the gut microbiome of Japanese is different from that of people in other countries, including Western and even other Asian countries, such as China. The Japanese gut microbiome has a greater abundance of taxa from the Actinobacteria phylum, particularly bacteria from the genus *Bifidobacterium*, than that in other nations [21]. Interestingly, the gut microbiome also differs among people living in different regions, even within the same country. The gut microbiota of 7009 individuals from 14 districts within one province in China was analyzed, and the individual’s geographic location showed an association with microbiota variation. Furthermore, microbiota-based metabolic disease models developed in one location failed when used elsewhere [22]. Japanese people have a unique dietary culture and habits compared to Western people, and given that individuals generally have a lower BMI and longer lifespans, one can surmise that these factors are related. Some Japanese people consume Western-style diets as well as traditional Japanese food, thus this group of individuals with varied Japanese lifestyles could serve as a good model to analyze the association between lifestyle, gut microbiome, and diseases. Additional factors can influence microbial composition such as exercise and physical activity. For instance, gut microbiomes from athletes also differ from those of normal populations. Athletes had a higher diversity of gut microbiota and significantly higher proportions of the genus *Akkermansia* compared with individuals with high BMI, indicating that exercise can affect the gut microbiome [23]. It should be noted that gut microbes interact with each other, and the overall picture of the microbiome cannot be determined by changes in single bacterial taxa. For example, obesity reduces SCFA-producing *Bifidobacteria*, but the total amount of SCFAs in the feces of obese people is increased [24]. This discrepancy suggests that SCFA production will be increased in gut microbiome of obese people by SCFA-producing taxa other than *Bifidobacteria.* In addition to the identification of the intestinal bacterial taxa, functional analysis of the microbiome and measurement of bacterial metabolites may also be useful in understanding the overall picture of the gut microbiota. Lifestyles affecting prostate cancer risk also change the gut microbiome (Table 1). Obesity, well-known risk factors of prostate cancer, decrease the ratio of *Firmicutes* to *Bacteroidetes* [25]. High-fat diets also affect gut microbiome, decreasing *Bacteroidetes* and increasing *Firmicutes* and *Proteobacteria* [9,26]. Dairy products increase prostate cancer risk, although it is still controversial [27]. Dairy products increase *Lactobacillus* and *Bifidobacterium* and decrease *Bacteroidetes* [28,29,30].

Gut dysbiosis impairs gut wall integrity, increases gut permeability, and reduces the expression of tight junction proteins such as zonula occludens (ZO)-1 and occludin, causing a “leaky gut.” A leaky gut results in the translocation of gut metabolites or bacterial components, such as lipopolysaccharide (LPS), into systemic circulation [31,32]. *Akkermansia muciniphilia* is involved in the maintenance of a healthy gut wall by degrading mucin, and its abundance is inversely associated with several diseases [33,34,35]. SCFAs are major metabolites of gut microbes, and butyrate plays a role in gut barrier function, and immunoregulations [36]. Obese individuals have higher levels of SCFAs in their feces compared to lean individuals [37], and the ratio of *Firmicutes* to *Bacteroidetes* is decreased in obese subjects. *Bifidobacterium*, known to improve the gut mucosal barrier and lower intestinal LPS levels, is reduced in the feces of obese people [25,38]. Interventions to improve dietary habits could change the gut microbiome. A randomized controlled study of obese individuals with metabolic syndrome showed that an energy-restricted Mediterranean diet and increased physical activity changed the gut microbiota. Changes in *Lachnospiraceae* were positively associated with adherence to the Mediterranean diet [39].
cancers-15-01375-t001_Table 1Table 1The lifestyles affecting prostate cancer risk and these effects in gut microbiome.Prostate Cancer RiskRisk FactorsChanges in Gut MicrobiotaReferencesHighObesityThe ratio of *Firmicutes* to
*Bacteroidetes* ↓*Bifidobacterium* ↓[24,38]High fat diet*Bacteroidetes*↓*Firmicutes* ↑*Proteobacteria* ↑[9,26]Dairy product*Lactobacillus* ↑*Bifidobacterium* ↑*Bacteroidetes*↓[28,29,30]LowMediterranean diet*Lachnospiaceae*↑[39]


## 3. The Gut Microbiome and Diseases

Gut microbes have direct contact with the intestinal wall; consequently, several intestinal diseases are affected by gut microbiota directly or indirectly through the modulation of local immune systems, such as regulatory T cell, dendric cells and CD4+ T cell [40]. In the gut microbiota of IBD patients, the abundance of specific bacteria, such as *Enterobacteriaceae* and *Fusobacteria*, are increased, indicating a direct association between gut microbes, local inflammation, and altered local host immunity [41,42]. Fecal microbiota transplantation, defined as the administration of fecal material containing distal gut microbiota from a healthy donor to the gastrointestinal tract of patients with IBD, has been conducted as a remedial treatment [43]. A systematic review showed that this approach may increase the rate of patients achieving clinical remission in ulcerative colitis (RR = 2.03) [43]. Apart from the gut, diseases of the liver and central nervous system are also reportedly affected by the gut microbiome. The liver receives bacterial components and metabolites that are absorbed in the intestinal tract and delivered via portal circulation [44]. Phenyl sulfate, a metabolite derived from intestinal bacteria, acts as an aggravating compound in diabetic kidney disease and contributes to albuminuria [45]. However, inhibition of sodium-glucose cotransporter 1 (SGLT1) in the gut ameliorates renal failure by altering the gut microbiome and reducing phenyl sulfate concentrations [46]. The increased influx of gut microbiota-derived endotoxins in portal circulation promotes TLR-4 expression and is associated with hepatic inflammation and steatosis [47,48].

Gut dysbiosis due to a HFD impairs the barrier of the intestinal wall, leading to the leakage of LPS into the systemic circulation. Systemic inflammation due to endotoxemia is implicated with the pathogenesis of insulin resistance and type 2 diabetes mellitus [49]. Endotoxins and amyloids from gram-negative bacteria can penetrate the blood-brain barrier and induce amyloid β aggregation and neuroinflammation in the central nervous system, suggesting that bacterial molecules and metabolites may be involved in the onset and progression of Alzheimer’s disease [50]. Gut microbiota-derived SCFAs promote neuroinflammation and tau-mediated neurodegeneration in the hippocampus [51]. These gut microbiome-mediated associations between gut and these distant organs are referred to as the “gut-brain axis” and “gut-liver axis” (Figure 1). However, these associations are not limited to these organs and are likely to include other systems.

## 4. The Gut Microbiome and Cancer

In recent years, it has become evident that gut microbiota affects various types of cancers. Multiple bacterial taxa and their metabolites have contributed to the development and progression of colorectal cancer [52]. Metagenomic and metabolomic studies on feces from participants who underwent colonoscopy showed that the relative abundance of *Fusobacterium nucleatum* was correlated with cancer progression [53]. *Fusobacterium nucleatum* has been reported in other studies as a colorectal cancer-associated bacterium that promotes tumor progression in a mouse model of intestinal cancer in a non-inflammatory manner [54]. In the liver, translocated bacterial metabolites and components may be involved in hepatocellular carcinoma (HCC). Obesity-induced hepatic translocation of lipoteichoic acid, a gram-positive intestinal bacterial component, accelerated senescence of hepatic stellate cells, promoting HCC progression through PGE_2_-mediated suppression of antitumor immunity [55]. It has been suggested that the gut microbiota may influence the development of breast cancer by deconjugating conjugated estrogen excreted in the intestinal tract, and thus allowing the biologically active form to be reabsorbed by the host [56]. In addition, intestinal bacteria can metabolize estrogen-like compounds, such as enterodiol and enterolactone, suggesting that gut microbiota plays a role in breast cancer development [57]. The focus on gut microbiota has not only been directed toward its impact on cancer development and progression, but also on the indirect effects in response to drug therapy, including immune checkpoint inhibitors (ICIs) [58]. In patients with advanced renal cell carcinoma (RCC) treated with anti-PD-L1 therapy, antibiotic use during treatment compared with no use was associated with an increased rate of primary progressive disease (75 vs. 22%, respectively) and shorter progression-free survival (1.9 vs. 7.4 months, respectively HR = 3.1), suggesting that antibiotic-induced dysbiosis reduced the clinical benefit from immune checkpoint inhibitors [59]. Metagenomic data of fecal samples from advanced RCC patients treated with nivolumab showed that the abundance of *Akkermancia muciniphila* and *Bacteroides salyersiae* was increased in responders, and that transplantation of these bacteria or feces from responder patients to RCC mouse models rescued responsiveness to anti-PD-1 plus anti-CTLA-4 treatment from mice colonized with the microbiota of non-responder patients [60]. The mechanism by which the gut microbiome influences ICI therapeutic efficacy is not fully understood; however, further studies will lead to innovative methods to enhance ICI treatments.

## 5. The Gut Microbiome and Prostate Cancer

The association between the gut microbiome and prostate cancer has been studied in human samples. In 2018, Golombas et al. examined the gut microbiome in 20 men with either benign prostatic disease or high-risk prostate cancer and reported that *Bacteriodes massiliensis* was abundant in patients with prostate cancer compared to controls [61]. Sfanos et al. compared the gut microbiome of patients with prostate cancer who received androgen deprivation therapy (ADT) to those of healthy volunteers and found that *Akkermansiia muciniphila* and *Ruminococcaceae* were increased in patients treated with ADT [62]. Liss et al. analyzed rectal swabs from 133 men who received prostate needle biopsy and showed that *Streptococcus* and *Bacteroides* were increased in patients with prostate cancer. Although 16S rRNA amplicon sequencing alone cannot specifically identify functional genes, it can be used with various computational tools to infer the functional metagenome [63]. Predicted metagenome analysis revealed that the folate and arginine pathways were down-regulated in the gut microbiome of men with prostate cancer therefore implicating these pathways with the pathogenesis of prostate cancer. *Bacteroides massiliensis* were also found to be increased in the gut microbiome from patients with prostate cancer by another study using a small cohort (8 men with benign prostate hypertrophy and 12 with high-risk prostate cancer) [61]. Matsushita et al. analyzed the gut microbiome of 152 men who underwent prostate biopsy and found that *Rikenellaceae*, *Alistipes*, and *Lachonospira* were increased in patients with high-risk prostate cancer [64]. The predictive value of these bacterial taxa was similar to serum PSA levels for the high-risk prostate cancer group. However, the index comprised from a profile of 18 bacteria taxa identified high-risk prostate cancer cases more precisely. The area under the curve (AUC) of reservoir operating characteristic curve analysis of this index for detecting high-risk prostate cancer was 0.85, while the AUC of serum PSA levels was 0.74. Functional pathway analysis showed that five metabolic pathways (starch and sucrose metabolism, phenylpropanoid biosynthesis, phenylalanine, tyrosine, and tryptophan biosynthesis, cyanoamino acid metabolism, and histidine metabolism) were positively associated with high-risk prostate cancer. The analysis of fecal microbiome of 23 patients with metastatic castration-resistant prostate cancer (CRPC) resistant to enzalutamide prior to treatment with anti-PD-1 showed that the responders have increased levels of *Streptococcus salivarius.* Interestingly, *Akkermansia muciniphila* levels were reduced in the fecal samples from responders [65].

The above-mentioned studies suggest that the gut microbiome influences prostate cancer and vice versa. Until recently, the precise mechanisms driving their interactions remained largely undiscovered, however, new findings have shed light on the subject (Figure 2). In 2021, Matsushita et al. reported that SCFAs originating from the gut microbiome promoted the growth of prostate tumors in a transgenic mouse model of prostate cancer [66]. A HFD is considered to be a risk factor for prostate cancer and is implicated in promoting prostate cancer growth by inducing local inflammation [5]. Antibiotics administered to prostate-specific conditional *Pten*-knockout mice fed a HFD suppressed prostate cancer growth. Interestingly, different types of antibiotics showed different effects on tumor growth—indicating that specific taxa are responsible for this phenomenon. For example, gentamycin suppressed HFD-induced prostate cancer growth, but neomycin did not. Microarray analysis of prostate tumors showed that IGF-1 was significantly down-regulated in HFD-fed *Pten*-knockout mice treated with antibiotics compared with HFD-fed *Pten*-knockout mice without antibiotics. Antibiotics also decreased the insulin-like growth factor 1 (IGF-1) levels in serum in wild-type HFD-fed mice. IGF-1 stimulated prostate cancer growth via the mitogen-activated protein kinase (MAPK) and phosphatidylinositol-3 kinase (PI3K) signaling pathways. Short chain fatty acids (SCFAs), such as butyrate, acetate, isobutyrate, and lactate are major metabolites of the gut microbiome, SCFAs play important roles in the regulation of intestinal immune cells with anti-inflammatory effects. SCFAs are also known to suppress colon cancer. Butyrate, the major compound of SCFAs suppress colonic carcinogenesis through this anti-inflammatory effect [67]. On the other hand, it also has a variety of other physiological effects, such as Wnt signal modulation, and promotes colorectal cancer, depending on its concentration [68,69]. However, in the *Pten*-knockout prostate cancer mouse model, the administration of antibiotics decreased SCFA levels in the gut. SCFAs from gut microbiota are known to stimulate IGF-1 production, leading to bone growth [70]. In the *Pten*-knockout prostate cancer mouse model, SCFAs also stimulated the production of IGF-1 in prostate tumors and other organs, such as liver, contributing to augmented prostate cancer growth. Notably, IGF-1 expression was upregulated in prostate cancer of severe obese patients compared with non-obese patients. Patients with benign prostatic hyperplasia (BPH) also have higher levels of fecal isobutyric acid and isovaleric compared to healthy individuals, and isobutyric acid, isovaleric acid, and isocproic acid were associated with the occurrence of metabolic syndrome in patients with BPH [71]. Furthermore, the Firmicutes/Bacteroidetes (F/B) ratio was significantly higher in Japanese men with enlarged prostates than in men with normal-size prostates [72]. SCFAs derived the gut microbiota may be a risk factor for prostate cancer and BPH by a similar mechanism. BPH is a very frequent age-related disease [73]. BPH, characterized by hyperplasia of the transition zone, is associated with inflammation, oxidative stress, and other several biological factors [74]. The relationships between different diets and the development of BPH has been discussed for some time [75]. HFD, in which 60% of kcal consists of lipids, promoted BPH in rat models, and activation of ERK1/2 was likely involved in this process [76]. Matsushita et al. reported that HFD also promoted prostate cancer growth via gut microbiome through a similar mechanism, suggesting that the connection between BPH and diet may be also mediated by the gut microbiome [66]. Fat consumption was reported to increase the incidence of BPH in humans. In a 7-year prospective study of 4770 subjects in The Prostate Cancer Prevention Trial, the highest fat intake group had a significantly increased hazard ratio by 31% compared with the lowest group [77].

It is important to note that the consumption of a HFD causes gut dysbiosis which leads to the development of a “leaky gut” by down-regulating tight junction molecules, such as ZO-1 [32,78]. Leaky gut leads to the leakage of bacterial components, such as lipopolysaccharide (LPS) or lipoteichoic acids (LTA), into systemic circulation. Leaked LPS and LTA in turn provoke systemic inflammation which have wide ranging implications including cancer promoting effects [79]. In prostate cancer, LPS activates mast cells via toll-like receptor 4 [78]. HFD-fed *Pten*-knockout mice also demonstrated upregulated levels of histamine decarboxylase (HDC), which plays a crucial role in histamine production. Fexofenadine, an H1 receptor blocker, suppressed the expression of inflammatory cytokines, such as Il-6, IL-10, IL-4, and IL-17, and reduced infiltration of MDSCs into prostate tumors. Furthermore, fexofenadine suppressed prostate cancer growth in HFD-fed mice. This upregulation in HDC levels is due to leaked LPS from dysbiotic gut microbiota of HFD-fed mice, and thus LPS administration to control diet-fed mice leaded HDC upregulation. In addition, inhibition of LPS suppressed the prostate cancer growth of HFD-fed mice. Like obese HFD-fed mice, severely obese patients with prostate cancer exhibited increased tumor-infiltrating mast cells. Gut dysbiosis could also be associated with drug-resistance in prostate cancer [78]. Antibiotics-induced dysbiosis, characterized by the enrichment of *Proteobacteria*, resulted in the elevation of tumor LPS due to an increase in gut permeability. Intratumoral elevation of LPS activated the NF-κB-IL6-STAT3 axis, leading to prostate cancer growth and docetaxel-resistance [80].

Androgen and its receptor have a pivotal role in the development of CRPC and these studies provide evidence that shows that the gut microbiome is intricately implicated and are further explored in the next section.

## 6. The Gut Microbiome in Androgenesis

ADT has been the gold standard for the treatment of prostate cancer for decades. However, patients eventually develop resistance and progress to CRPC which relies on minute levels of androgens for its growth by amplifying the androgen receptor [81]. Androgens are produced in the testis, adrenal glands, and in prostate tumors by cancer cells, but recent studies suggested that the gut microbiota also plays a role in androgenesis (Figure 3). In the analysis of 31 young Korean men aged 25–65 years using 16SrRNA sequencing, the abundance of *Acinetobacter*, *Dorea*, *Ruminococcus*, and *Megamonas* correlated significantly with serum testosterone levels [82]. In the analysis of the gut microbiota of 54 Japanese men aged 65 years or older, serum testosterone levels were positively correlated with the abundance of *Firmicutes* [83]. Several studies have also reported the relationship between the microbiome and polycystic ovarian syndrome (PCOS), a disease caused by high testosterone production in women [84,85]. Testosterone levels increased when feces from male mice were transplanted into female mice, suggesting that specific intestinal bacteria promoted testosterone production. In addition, *Firmicutes* synthesize testosterone or promotes its reabsorption through unconjugation [86]. Gut microbiota modulate the enterohepatic circulation of androgens, affecting systemic androgen levels. Gut bacteria can also produce androgens from glucocorticoids [86]. ADT also affects the gut microbiome in both mice and humans and promotes the expansion of specific bacteria [87]. ADT plus abiraterone acetate depletes androgen-utilizing and pro-inflammatory *Corynebacterium* and increases *Akkermansia muciniphila* in the gut. Predicted metagenomic analysis from 16S rRNA sequencing suggested that patients with abiraterone acetate increased bacterial biosynthesis of vitamin K2, which is known to be an inhibitor of androgen-dependent and-independent tumor growth [88]. *Ruminococcus* has genes sharing high sequence homology with human *CYP17* and *Ruminococcus can* convert pregnenolone and hydroxypregnenolone in feces to DHEA and testosterone. Patients with CRPC had a higher abundance of *Ruminococcus* in their feces and promoted prostate cancer growth via the production of androgens in the gut. Abiraterone acetate, a selective inhibitor of CYP17A1, inhibited the bacterial conversion of pregnenolone to DHEA and testosterone. In contrast, FMT with hormone-sensitive microbiota or administration of *Prevotella stercorea* can decrease androgens levels in CTX mice and delay the onset of CRPC [87]. In this context, androgenesis by gut microbiota should be considered for patients undergoing ADT for metastatic hormone-sensitive prostate cancer or CRPC.

## 7. The Association with the Epigenetics

Prostate cancer pathobiology is affected not only by genomic mutation, but also by epigenetic modification, namely the acquired regulation of gene expression [89]. Various environmental factors can cause alterations in the epigenome and may be drivers of cancer formation and progression [90]. Aberrant DNA hypermethylation is a prevalent epigenetic modification responsible for the inactivation of tumor suppressor genes in prostate cancer, and *GSTP1*, a class of Glutathione S-transferases (GSTs), a family of enzymes responsible for processes that protect cells from xenobiotics, was earliest reported to be hypermethylated in human prostate cancer [91]. Similarly, DNA hypermethylation is involved in the expression not only of DNA repair genes but also of the genes involved in cell cycle, apoptosis, and cell adhesion, which have been attempted to be regulated by dietary therapy [92]. The methyl group is extracted from S-adenyl methionine, and bacterial metabolites such as folate and betaine are essential for its synthesis [93]. Certain strains of *Lactobacillus* and *Bifidobacterium* used as probiotics have the ability to generate folate, and such strains may enhance prostate cancer risk via DNA hypermethylation [94]. Contrarily, in a gene-based predictive functional analysis of the gut microbiota by Liss et al. the function to synthesize folate was significantly reduced in prostate cancer patients compared to men without cancer [63]. It is possible that the functional analysis is based on gene prediction and does not reflect the actual folate synthesis capacity, but further studies are needed to address this discrepancy.

Epigenetic modifications of histones, which give the backbone to chromatin, have been reported in prostate cancer [89]. One type of histone modification is methylation, which may also be influenced by gut microbiota-derived metabolites involved in methyl group donation. The other type, histone acetylation, leads to chromatin loosening, leading to transcription activation. Histone deacetylases (HDACs), which act in transcription inactivation by clearing acetyl groups, are inhibited by short-chain fatty acids (SCFAs) produced by some anaerobic bacteria fermenting dietary fiber [95]. Butyrate, one type of SCFA, at high concentrations, inhibited prostate cancer growth in vitro by altering the expression of cell cycle regulators and AR through the epigenetic histone modification [96]. However, butyrate has contrasting effects on cancer cells depending on its concentration, and we have shown that deficiency of gut microbiota-derived SCFAs rather inhibits prostate cancer growth in vivo [66,97]. These findings suggest that the gut microbiota may also be involved in epigenetic modifications of prostate cancer cells, although these associations have not yet been directly demonstrated. Future studies are needed.

## 8. Limitations

In these human gut microbiota analyses, the proportion of several intestinal bacteria seems to change according to the prostate cancer status of the host. Most of the human-reported studies have been conducted in the limited regions of Asia and the United States, although the composition of the gut microbiota varies between regions due to the diversity of lifestyles such as dietary habit. In other to achieve the identification of intestinal bacteria that truly work as promotive or preventive factors of prostate cancer linked to lifestyles, extensive global microbiota research of prostate cancer patients will be necessary.

## 9. Conclusions

The gut microbiome is greatly influenced by several environmental factors, such as lifestyle. Change in the gut microbiota can be involved in prostate cancer progression through its metabolites and endotoxins. A greater understanding of the molecular mechanisms underlying these bidirectional interactions have allowed us to establish a “gut-prostate-axis”. These interventions could be further incorporated with current treatments as novel strategies for the prevention and management of human prostate cancer.

## Figures and Tables

**Figure 1 cancers-15-01375-f001:**
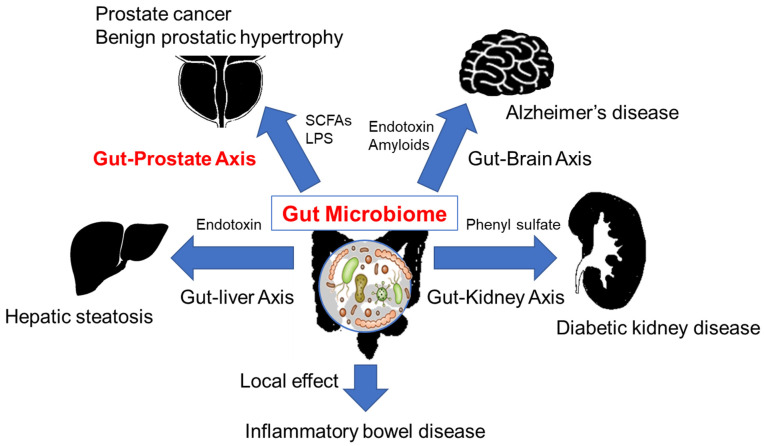
Local effect and gut-distant organ axis via gut microbiome. The endotoxins and the metabolite from the gut microbiome affect the distant organs, SCFA: short-chain fatty acid, LPS: lipopolysaccharides.

**Figure 2 cancers-15-01375-f002:**
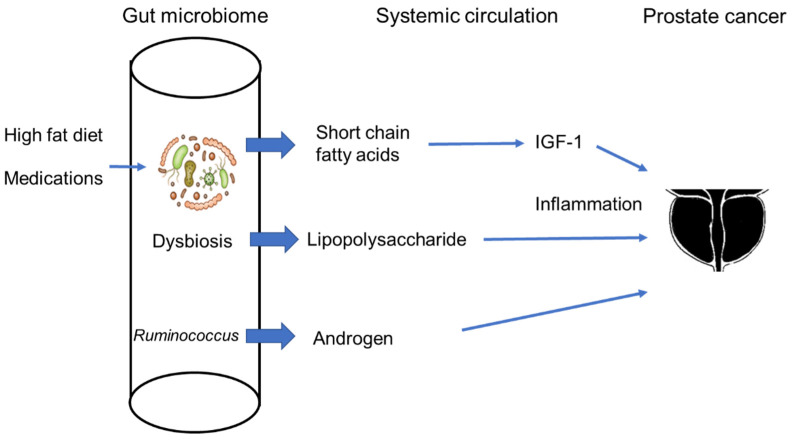
Gut-prostate axis. Diet and medications affect gut microbiome. Gut dysbiosis results in the leakage of endotoxins into systemic circulation. Gut microbiota also produce androgen affecting prostate cancer progression.

**Figure 3 cancers-15-01375-f003:**
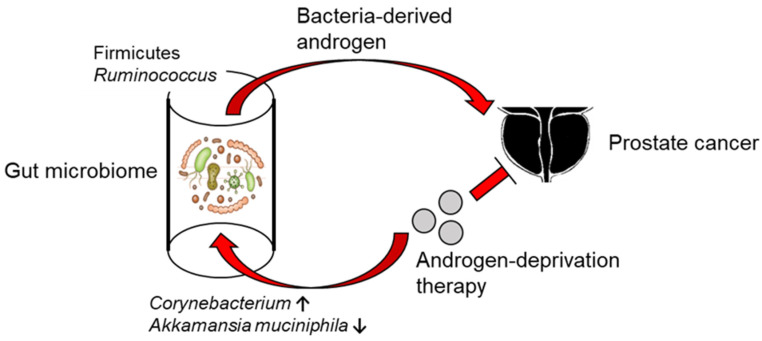
Interactions between the gut microbiome and prostate cancer in androgen regulation. Gut microbiome-derived androgen may be involved in prostate cancer growth. Androgen-deprivation therapy conversely change the composition of the gut microbiota.

## Data Availability

Not applicable.

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
