# Peer review of "The Gut-Prostate Axis: A New Perspective of Prostate Cancer Biology through the Gut Microbiome"

_cancers, 2023, doi:10.3390/cancers15051375_

Round 1

Reviewer 1 Report

The authors delve upon the gut microbiome/dysbiosis influencing prostate carcinogenesis/mCRPC and its progression. They give an overview of molecular mechanisms associated with these and subtly discuss the interventions. The manuscript is written well with some well documented references.

A few suggestions:

Ockzowski et al and Fang et al. 2019 have worked on nutrition, dietetics and metatranscriptomics/metagenomic works respectively which the authors can cite their papers.

In the first paragraph, the authors oculd lay emphaiss on the lifetsyle an dtype of diet patterns ( non vegetarian/red meat etc) which could lead to prostate/gut dysbiosis and ultimately the mCRPC

A table of these patterns in lieu of gut dysbiosis and diet patterns would be an excellent insertion

A paragraph on epigenetics and environment influence will be nice

The Benign Prostatic Hyperplasia (BPH) and diet patterns along with the aging factors may be kindly added in a word or two. How BPH is caused due to aging etc needs a mention

Pl find attachment for subtle suggestions on language

Scores on a scale of 0-5 with 5 being the best

Language: 4

Novelty: 4

Brevity: 4

Scope and relevance: 4.5

Author Response

Reviewer 1

The authors delve upon the gut microbiome/dysbiosis influencing prostate carcinogenesis/mCRPC and its progression. They give an overview of molecular mechanisms associated with these and subtly discuss the interventions. The manuscript is written well with some well documented references.

Response: Thank you for your comment.

A few suggestions:

Ockzowski et al and Fang et al. 2019 have worked on nutrition, dietetics and metatranscriptomics/metagenomic works respectively which the authors can cite their papers.

In the first paragraph, the authors oculd lay emphaiss on the lifetsyle an dtype of diet patterns ( non vegetarian/red meat etc) which could lead to prostate/gut dysbiosis and ultimately the mCRPC

A table of these patterns in lieu of gut dysbiosis and diet patterns would be an excellent insertion

Thank you for your suggestions. However, unfortunately we could not find these paper you suggested. Instead, we added the following table and the sentences.

Lifestyles affecting prostate cancer risk also change the gut microbiome (Table 1). Obesity, well-known risk factors of prostate cancer, decrease the ratio of Firmicutes to Bacteroidetes[25]. High-fat diet also affect gut microbiome, decreasing Bacteroidetes and in-creasing Firmicutes and Proteobacteria[9,26]. Dairy product increasing prostate cancer risk, although it is still controversial[27] . Dairy product increases Lactobacillus and Bifidobacte-rium and decreases Bacteroidetes [28–30]  

Table 1. The lifestyles affecting prostate cancer risk and these effects in gut microbiome.

Prostate cancer risk

Risk factors

Changes in gut microbiota

References

High

Obesity

The ratio of Firmicutes to Bacteroidetes

Bifidobacterium

[24] [38]

High fat diet

Bacteroidetes

Firmicutes

Proteobacteria

[9], [26]

Dairy product

Lactobacillus

Bifidobacterium

Bacteroidetes

[28], [29], [30]

Low

Mediterranean diet

Lachnospiaceae

[39]

A paragraph on epigenetics and environment influence will be nice

Ans. We added the following sentences according to the reviewer’s suggestion.

  1. The association with the epigenetics

Prostate cancer pathobiology is affected not only by genomic mutation, but also by epigenetic modification, namely the acquired regulation of gene expression [89]. Various environmental factors can cause alterations in the epigenome and may be drivers of cancer formation and progression[90]. Aberrant DNA hypermethylation is a prevalent epigenetic modification responsible for the inactivation of tumor suppressor genes in prostate cancer, and GSTP1, a class of Glutathione S-transferases (GSTs), a family of enzymes responsible for processes that protect cells from xenobiotics, was earliest reported to be hypermethylated in human prostate cancer[91]. Similarly, DNA hypermethylation is involved in the expression not only of DNA repair genes but also of the genes involved in cell cycle, apoptosis and cell adhesion, which have been attempted to be regulated by dietary therapy [92]. The methyl group is extracted from S-adenyl methionine, and bacterial metabolites such as folate and betaine are essential for its synthesis [93]. Certain strains of Lactobacillus and Bifidobacterium used as probiotics have the ability to generate folate, and such strains with may enhance prostate cancer risk via DNA hypermethylation[94]. Contrarily, in a gene-based predictive functional analysis of the gut microbiota by Liss et al. the function to synthesize folate was significantly reduced in prostate cancer patients compared to men without cancer[63]. It is possible that the functional analysis is based on gene prediction and does not reflect the actual folate synthesis capacity, but further studies are needed to address this discrepancy.

Epigenetic modifications of histones, which give the backbone to chromatin, have been reported in prostate cancer[89]. One type of histone modification is methylation, which may also be influenced by gut microbiota-derived metabolites involved in methyl group donation. The other type, histone acetylation, leads to chromatin loosening, leading to transcription activation. Histone deacetylases (HDACs), which act in transcription inactivation by clearing acetyl groups, are inhibited by short-chain fatty acids (SCFAs) produced by some anaerobic bacteria fermenting dietary fiber[95]. Butyrate, one type of SCFAs, at high concentrations, inhibited prostate cancer growth in vitro by altering the expression of cell cycle regulators and AR through the epigenetic histone modification [96]. However, butyrate has contrasting effects on cancer cells depending on its concentration, and we have shown that deficiency of gut microbiota-derived SCFAs rather inhibits prostate cancer growth in vivo[66,97]. These findings suggest that the gut microbiota may also be involved in epigenetic modifications of prostate cancer cells, although these associations have not yet been directly demonstrated. Future studies are needed.

The Benign Prostatic Hyperplasia (BPH) and diet patterns along with the aging factors may be kindly added in a word or two. How BPH is caused due to aging etc needs a mention

Response:Thank you for the meaningful advice. We added the following sentences.

BPH is a very frequent age-related disease [73]. BPH, characterized by hyperplasia of the transition zone, is associated with inflammation, oxidative stress, and other several biological factors [74]. The relationships between different diets and the development of BPH has been discussed for some time [75]. HFD, in which 60% of kcal consists of lipids, promoted BPH in rat models, and activation of ERK1/2 was likely involved in this process [76]. Matsushita et al. reported that HFD also promoted prostate cancer growth via gut microbiome through a similar mechanism, suggesting that the connection between BPH and diet may be also mediated by the gut microbiome[66]. Fat consumption was reported to increase the incidence of BPH in humans. In a 7-year prospective study of 4770 subjects in The Prostate Cancer Prevention Trial, the highest fat intake group had a significantly increased hazard ratio by 31% compared with the lowest group [77].

Reviewer 2 Report

-       A brief summary

In this review, Fujita et al., provide an overview of the current relationship between gut microbiota and prostate cancer known as the “gut-prostate axis”. The authors reviewed some corelative studies from human subjects that suggested a potential relationship between gut microbes and prostate cancer development and progression. They also highlighted how a high-fat diet and gut microbiota dysbiosis could promote tumorigenesis in the prostate cancer mouse model. They discussed some of the underlying mechanisms by which gut microbiota dysbiosis promotes prostate cancer via the production of microbial metabolites or by promoting systemic inflammation. They also shed light on the impact of prostate cancer treatment, particularly androgen deprivation therapy on the gut microbiota, and discussed recent data indicating that gut microbiota could promote castration-resistant prostate cancer via the production of androgen. This manuscript is very similar to a review published earlier on the same topic: Gut microbiome and prostate cancer (PMID 35388531).

-       Overall recommendations:

All in all, a significant rewrite is needed to provide an original and unbiased review of the current knowledge regarding gut microbes and prostate cancer.

-       General comments

Overall, the manuscript is well written with very few typos that are listed below. The topic of this review is relevant and sheds light on the involvement of gut microbes in the tumorigenesis of prostate cancer. The most recent evidence included in this manuscript has shown a causal link between gut microbes and prostate cancer using preclinical models of prostate cancer. However, given the number of recent publications covering the area of gut microbiota and prostate cancer, I find it odd that the authors put much of their focus on a few select publications to detail the importance of gut microbes in prostate cancer. Although the manuscript was well-structured, there was a lack of highlighting studies’s limitations such as the fact that most human-reported studies on prostate cancer and gut microbiota were conducted in Asia and the US. Since diet is a key modulator of gut microbiota and varies greatly across the globe, the authors should discuss the lack of knowledge about the gut microbiota composition of prostate cancer patients in other parts of the world.

The main issue of this review by Fujita et al., is with the similarity with a review on the same topic in 2022 (see reference Fujita et al., 2022 Journal of Urology). Unless the authors broaden their discussions and future directions, regarding the gut microbiota-prostate cancer axis and the underlying mechanisms as a reviewer, I do not think that the content of this review is justified.

Specific points:

Figures:

I do not find figure 1 informative. The caption states: “Gut microbiome-mediated associations between gut and distant organs”, however the authors do not describe any of these associations and instead make plentiful use of the term “axis”. If this representation is to be included in the manuscript I suggest to provide a fully annotated and comprehensive figure describing potential mechanisms of action.

Figure 2 would also benefit from a makeover. It is not clear why “High fat diet”, “Antibiotic” and “Androgen deprivation therapy” are left unconnected to the rest of the graphical representation.  Also, is it really a dysbiotic microbiota when for example the authors describe studies that found 1 taxa difference between BPH and PCa fecal samples? There were no mention of studies describing the impact of antibiotics on clinical PCa, this information stem from the study of Matsushita et al., therefore, the authors need to specify that Figure 2 related to a mice model of PCa.

Main text:

The authors stated that prostate cancer-related mortality was increasing in the United States and Japan, but no statistics were provided. The authors should provide or reference recent mortality or incidence trend data that support their claims.

The statement: “It should be noted that gut microbes interact with each other, and the overall picture of the microbiome cannot be determined by changes in single bacterial taxa. For example, obesity reduces SCFA-producing Bifidobacteria, but the total amount of SCFAs in the feces of obese people is increased [24].” Needs to be clarified and in the context of the current manuscript it suggests that 16S data is unreliable.

Lines 141-148; Is this relevant to the review’s topic? Also, references 24-27 are reviews, please provide the corresponding publications of the original work.

Lines 169-178; the gut-brain axis relates to the enteric nervous system rather than amyloid aggregation, please rewrite this section.

Lines 251-264: Is it all related to Matsushita et al.,? if not, please provide the references. Improved clarity

The authors extensively rely on data published by Matsushita et al., this is especially problematic when the cited work is unrelated to the microbiota. In another instance the authors cite work by Matsushita et al., to introduce the general concept of gut permeability, tight junction dynamics, LPS and LTA-mediated systemic inflammation. In the current form, the review gives the impression of showcasing the work of a specific research group.

Line 264: The statement: “SCFAs are also known to suppress colon cancer” is wrong please review the literature there are plenty of example whereby SCFA can also promote colorectal cancer.

Line 275-277: “It may be possible that similar mechanisms could be involved in the development of prostate cancer and BPH”., please clarify.

Line 281-286: Please provide the appropriate references, not Matsushita et al.,

Lines 287-293: This is not relevant to the gut microbiota, consider removing.

The authors make several claims about the work by Yufei Lui et al., that are not accurate. First, this set of data is problematic since it lacks a no FMT group, to control for the normal TRAMP tumor development or the mitigating effect of antibiotic treatments on TRAMP tumors.  Therefore, it is impossible to conclude on the tumor suppressor or tumor promoting action of the human PCa microbiota. Second, the experimental design consisted of FMTs twice weekly for 12 weeks. It might not represent an FMT transfer and microbiota colonisation experiment but rather a continuous exposure to human fecal samples, therefore, these evidences should be presented in the context of the original experiments. Alternatively, the authors might want to consider removing this section since it is redundant with findings by Pernigoni et al., that are described later by the authors.

Lines 310-311; This sentence is to general, please remove.

Pernigoni et al., did not report reduced TRAMP-C1 tumor growth with prevotella stecorea this is must be corrected.

The authors are describing chicken and egg type of data, with sex hormones driving changes in the microbiota and gut microbes differentially impacting testosterone levels. Please provide a comprehensive assessment of the dynamics between hormones and bacteria and possibly include a figure to help with the general understanding of it.

Conclusion:

There are many issues with the conclusion that require significant rewording.

Why is “Conclusions” plural?

The statement: “The gut microbiome is greatly influenced not only by our genetic makeup” is not factual. The current data rather support the opposite, namely that the greater proportion of gut microbial populational changes is influenced by the environment.

What do the authors mean by “lifestyle choices”?

The authors state that: “Dysbiosis of the gut alters microbial composition which can in turn promote prostate carcinogenesis and progression.” I am not convinced that dysbiosis apply to microbiota changes reported in prostate cancer patients thus far and there is no convincing data as of yet, on prostate carcinogenesis with regard to a possible causal role of the gut microbes.

Prebiotics and probiotics were never introduced or discussed in the main text, please describe in the text or remove from the conclusion.

Minor points:

Line 12 and 61, present tense for: “hypothesized”

Line 39, typo: “Prostate cancer develops due the mutation”

Line 118: consider changing “host location” to “individual’s geographic location”

Line 135 and 138: weird characters

Line 152, what are “local immune systems” referring to?

Line 352: why thanking the laboratory and hospital staff for collecting, storing and managing samples?

Author Response

Reviewer 2

In this review, Fujita et al., provide an overview of the current relationship between gut microbiota and prostate cancer known as the “gut-prostate axis”. The authors reviewed some corelative studies from human subjects that suggested a potential relationship between gut microbes and prostate cancer development and progression. They also highlighted how a high-fat diet and gut microbiota dysbiosis could promote tumorigenesis in the prostate cancer mouse model. They discussed some of the underlying mechanisms by which gut microbiota dysbiosis promotes prostate cancer via the production of microbial metabolites or by promoting systemic inflammation. They also shed light on the impact of prostate cancer treatment, particularly androgen deprivation therapy on the gut microbiota, and discussed recent data indicating that gut microbiota could promote castration-resistant prostate cancer via the production of androgen. This manuscript is very similar to a review published earlier on the same topic: Gut microbiome and prostate cancer (PMID 35388531).

Response: We appreciate your valuable comment.

Overall recommendations:

All in all, a significant rewrite is needed to provide an original and unbiased review of the current knowledge regarding gut microbes and prostate cancer.

General comments

Overall, the manuscript is well written with very few typos that are listed below. The topic of this review is relevant and sheds light on the involvement of gut microbes in the tumorigenesis of prostate cancer. The most recent evidence included in this manuscript has shown a causal link between gut microbes and prostate cancer using preclinical models of prostate cancer. However, given the number of recent publications covering the area of gut microbiota and prostate cancer, I find it odd that the authors put much of their focus on a few select publications to detail the importance of gut microbes in prostate cancer. Although the manuscript was well-structured, there was a lack of highlighting studies’s limitations such as the fact that most human-reported studies on prostate cancer and gut microbiota were conducted in Asia and the US. Since diet is a key modulator of gut microbiota and varies greatly across the globe, the authors should discuss the lack of knowledge about the gut microbiota composition of prostate cancer patients in other parts of the world.

Response;

Thank you for the comments. Several papers other than those by Matsushita et al. were newly cited in this revised manuscript. These additional papers may provide fairer and stronger indications of a microbiota-mediated gut-prostate axis. We also mentioned the limitation that most of human-reported studies have been conducted in a few regions.

In these human gut microbiota analyses, the proportion of several intestinal bacteria seems to change according to the prostate cancer status of host. Most of human-reported studies has been conducted in the limited regions of Asia and the United States, although the composition of the gut microbiota varies between regions due to the diversity of lifestyles such as dietary habit. In other to achieve the identification of intestinal bacteria that truly work as promotive or preventive factors of prostate cancer linked to lifestyles, extensive global microbiota research of prostate cancer patients will be necessary.

The main issue of this review by Fujita et al., is with the similarity with a review on the same topic in 2022 (see reference Fujita et al., 2022 Journal of Urology). Unless the authors broaden their discussions and future directions, regarding the gut microbiota-prostate cancer axis and the underlying mechanisms as a reviewer, I do not think that the content of this review is justified.

Response;

We appreciate your valuable opinion. We added the new paragraph about the associations of gut microbiome with the epigenetic change in prostate cancer cells as follows;

Prostate cancer pathobiology is affected not only by genomic mutation, but also by epigenetic modification, namely the acquired regulation of gene expression [89]. Various environmental factors can cause alterations in the epigenome and may be drivers of cancer formation and progression[90]. Aberrant DNA hypermethylation is a prevalent epigenetic modification responsible for the inactivation of tumor suppressor genes in prostate cancer, and GSTP1, a class of Glutathione S-transferases (GSTs), a family of enzymes responsible for processes that protect cells from xenobiotics, was earliest reported to be hypermethylated in human prostate cancer[91]. Similarly, DNA hypermethylation is involved in the expression not only of DNA repair genes but also of the genes involved in cell cycle, apoptosis and cell adhesion, which have been attempted to be regulated by dietary therapy [92]. The methyl group is extracted from S-adenyl methionine, and bacterial metabolites such as folate and betaine are essential for its synthesis [93]. Certain strains of Lactobacillus and Bifidobacterium used as probiotics have the ability to generate folate, and such strains with may enhance prostate cancer risk via DNA hypermethylation[94]. Contrarily, in a gene-based predictive functional analysis of the gut microbiota by Liss et al. the function to synthesize folate was significantly reduced in prostate cancer patients compared to men without cancer[63]. It is possible that the functional analysis is based on gene prediction and does not reflect the actual folate synthesis capacity, but further studies are needed to address this discrepancy.

Epigenetic modifications of histones, which give the backbone to chromatin, have been reported in prostate cancer[89]. One type of histone modification is methylation, which may also be influenced by gut microbiota-derived metabolites involved in methyl group donation. The other type, histone acetylation, leads to chromatin loosening, leading to transcription activation. Histone deacetylases (HDACs), which act in transcription inactivation by clearing acetyl groups, are inhibited by short-chain fatty acids (SCFAs) produced by some anaerobic bacteria fermenting dietary fiber[95]. Butyrate, one type of SCFAs, at high concentrations, inhibited prostate cancer growth in vitro by altering the expression of cell cycle regulators and AR through the epigenetic histone modification [96]. However, butyrate has contrasting effects on cancer cells depending on its concentration, and we have shown that deficiency of gut microbiota-derived SCFAs rather inhibits prostate cancer growth in vivo[66,97]. These findings suggest that the gut microbiota may also be involved in epigenetic modifications of prostate cancer cells, although these associations have not yet been directly demonstrated. Future studies are needed.

Specific points:

Figures:

I do not find figure 1 informative. The caption states: “Gut microbiome-mediated associations between gut and distant organs”, however the authors do not describe any of these associations and instead make plentiful use of the term “axis”. If this representation is to be included in the manuscript I suggest to provide a fully annotated and comprehensive figure describing potential mechanisms of action.

Ans.: We appreciate your suggestion. We changed Figure 1 and its caption to include the word “axis” and the mechanisms as following.

Figure 1 Legend

Local effect and gut-distant organ axis via gut microbiome. The endotoxins and the metabolite from gut microbiome affect the distant organs, SCFA: short-chain fatty acid, LPS: lipopolysaccharides.

Figure 2 would also benefit from a makeover. It is not clear why “High fat diet”, “Antibiotic” and “Androgen deprivation therapy” are left unconnected to the rest of the graphical representation.  Also, is it really a dysbiotic microbiota when for example the authors describe studies that found 1 taxa difference between BPH and PCa fecal samples? There were no mention of studies describing the impact of antibiotics on clinical PCa, this information stem from the study of Matsushita et al., therefore, the authors need to specify that Figure 2 related to a mice model of PCa.

Ans.: We agree with your points. The animal study of Matsushita et al. proved that gut dysbiosis lead to leakage of LPS and to progression of prostate cancer. We revised Figure 2 to more clearly show the relationship between environmental factors, gut microbiome, and the identified mechanism involved in prostate cancer. We also changed the caption.

Figure 2. Gut-prostate axis. Diet and medications affect gut microbiome. Gut dysbiosis results in the leakage of endotoxins into systemic circulation. Gut microbiota also produce androgen affecting prostate cancer progression.

Main text:

The authors stated that prostate cancer-related mortality was increasing in the United States and Japan, but no statistics were provided. The authors should provide or reference recent mortality or incidence trend data that support their claims.

Ans, Thank you for the meaningful comments. We a corrected the main documents as following.

The incidence of advanced prostate cancer has been increasing in the USA and Japan, although the incidence of all prostate cancer in the USA has decreased since the recommendation of the US preventive service task force for prostate-specific antigen (PSA) screening in 2012[1].

The statement: “It should be noted that gut microbes interact with each other, and the overall picture of the microbiome cannot be determined by changes in single bacterial taxa. For example, obesity reduces SCFA-producing Bifidobacteria, but the total amount of SCFAs in the feces of obese people is increased [24].” Needs to be clarified and in the context of the current manuscript it suggests that 16S data is unreliable.

Ans.: Our point this statement is not that 16S data is unreliable, but that in addition to the identification of the intestinal bacterial taxa, functional analysis of the microbiome and measurement of bacterial metabolites may also be useful in understanding the overall picture of the gut microbiota. This statement was revised to make our point clear.

Main text:

It should be noted that gut microbes interact with each other, and the overall picture of the microbiome cannot be determined by changes in single bacterial taxa. For example, obesity reduces SCFA-producing Bifidobacteria, but the total amount of SCFAs in the feces of obese people is increased[24]. This discrepancy suggests that SCFA production will be increased in gut microbiome of obese people by SCFA-producing taxa other than Bifidobacteria. In addition to the identification of the intestinal bacterial taxa, functional analysis of the microbiome and measurement of bacterial metabolites may also be useful in understanding the overall picture of the gut microbiota.

Lines 141-148; Is this relevant to the review’s topic? Also, references 24-27 are reviews, please provide the corresponding publications of the original work.

Ans.: We showed the association of lifestyles which are known to change the risk of prostate cancer with gut microbiome. References 24-27 were changed to their original works.

Lines 169-178; the gut-brain axis relates to the enteric nervous system rather than amyloid aggregation, please rewrite this section.

Ans.: We highlighted in these lines the impact of the gut microbiota on remote organs. Therefore, we described the relationship with the brain rather than with the local enteric nervous system. To further confirm the axis between gut microbiota and brain, we newly cited the recently reported study (PMID36634180) in Science showing that gut bacterial metabolite, SCFAs, contribute to neuroinflammation, and tau-mediated neurodegeneration in the central nervous system.

Main text:

Gut dysbiosis due to a HFD impairs the barrier of intestinal wall, leading to the leak-age of LPS into the systemic circulation. Systemic inflammation due to endotoxemia is implicated with the pathogenesis of insulin resistance and type 2 diabetes mellitus[39]. Endotoxins and amyloids from gram-negative bacteria can penetrate the blood-brain barrier and induce amyloid β aggregation and neuroinflammation in the central nervous system, suggesting that bacterial molecules and metabolites may be involved in the onset and progression of Alzheimer’s disease[40]. Gut microbiota-derived SCFAs promote neuroinflammation and tau-mediated neurodegeneration in the hippocampus[51].. These gut microbiome-mediated associations between gut and these distant organs are referred to as the “gut-brain axis” and “gut-liver axis”. (Figure 1) However, these associations are not limited to these organs and are likely to include other systems.

Lines 251-264: Is it all related to Matsushita et al.,? if not, please provide the references. Improved clarity

Ans: Thank you for your point. The contents of these lines all relate to reference 56 of Matsushita et al.

The authors extensively rely on data published by Matsushita et al., this is especially problematic when the cited work is unrelated to the microbiota. In another instance the authors cite work by Matsushita et al., to introduce the general concept of gut permeability, tight junction dynamics, LPS and LTA-mediated systemic inflammation. In the current form, the review gives the impression of showcasing the work of a specific research group.

Ans: We appreciate your opinions. Research about gut-prostate axis is still undeveloped, especially in basic research, but we have tried to review the works as evenly as possible. We changed the citation to the appropriate one in the general concept of gut permeability, tight junction dynamics, LPS and LTA-mediated systemic inflammation.

Line 264: The statement: “SCFAs are also known to suppress colon cancer” is wrong please review the literature there are plenty of example whereby SCFA can also promote colorectal cancer.

Ans: We appreciate the meaningful suggestion. SCFAs, have several biological effects besides its anti-inflammatory effect and works against colorectal cancer, both promotive and inhibitory. This knowledge has been newly described in our revised manuscript.

Main text (line264)

Butyrate, the major compound of SCFAs suppress colonic carcinogenesis through this anti-inflammatory effect [67]. On the other hand, it also has a variety of other physiological effects, such as Wnt signal modulation, and promotes colorectal cancer, depending on its concentration.[68,69]

Line 275-277: “It may be possible that similar mechanisms could be involved in the development of prostate cancer and BPH”., please clarify.

Ans: Thank you for your opinion. We revised this sentence to be clearer.

Main text (line275-277)

SCFAs derived the gut microbiota may be a risk factor for prostate cancer and BPH by a similar mechanism.

Line 281-286: Please provide the appropriate references, not Matsushita et al.,

Ans: We changed the reference to the appropriate one.

Main text (line281-286)

It is important to note that the consumption of a HFD causes gut dysbiosis which leads to the development of a “leaky gut” by down-regulating tight junction molecules, such as ZO-1 [32][78]. Leaky gut leads to the leakage of bacterial components, such as lipo-polysaccharide (LPS) or lipoteichoic acids (LTA), into systemic circulation. Leaked LPS and LTA in turn provoke systemic inflammation which have wide ranging implications including cancer promoting effects[79].

Lines 287-293: This is not relevant to the gut microbiota, consider removing.

Ans: Thank you for your thoughtful suggestion. We added sentences to the lines you mentioned to show that these are relevant to the gut microbiota.

Main text (Line287-293)

It is important to note that the consumption of a HFD causes gut dysbiosis which leads to the development of a “leaky gut” by down-regulating tight junction molecules, such as ZO-1.[32][78] Leaky gut leads to the leakage of bacterial components, such as lip-opolysaccharide (LPS) or lipoteichoic acids (LTA), into systemic circulation. Leaked LPS and LTA in turn provoke systemic inflammation which have wide ranging implications including cancer promoting effects[79]. In prostate cancer, LPS activates mast cells via toll-like receptor 4.[78] HFD-fed Pten-knockout mice also demonstrated upregulated levels of histamine decarboxylase (HDC), which plays crucial roles in histamine production. Fexofenadine, an H1 receptor blocker, suppressed the expression of inflammatory cyto-kines, such as Il-6, IL-10, IL-4, and IL-17, and reduced infiltration of MDSCs into prostate tumors. Furthermore, fexofenadine suppressed prostate cancer growth in HFD-fed mice. This upregulation in HDC levels is due to leaked LPS from dysbiotic gut microbiota of HFD-fed mice, and thus LPS administration to control diet-fed mice leaded HDC upregu-lation. In addition, inhibition of LPS suppressed the prostate cancer growth of HFD-fed mice. Like obese HFD-fed mice, severely obese patients with prostate cancer exhibited in-creased tumor-infiltrating mast cells. Gut dysbiosis could be also associated with drug-resistance in prostate cancer[78]. Antibiotics-induced dysbiosis, characterized by the enrichment of Proteobacteria, resulted in the elevation of tumor LPS due to an increase in gut permeability. Intratumoral elevation of LPS activated the NF-κB-IL6-STAT3 axis, leading to prostate cancer growth and docetaxel-resistance.[80]

The authors make several claims about the work by Yufei Lui et al., that are not accurate. First, this set of data is problematic since it lacks a no FMT group, to control for the normal TRAMP tumor development or the mitigating effect of antibiotic treatments on TRAMP tumors.  Therefore, it is impossible to conclude on the tumor suppressor or tumor promoting action of the human PCa microbiota. Second, the experimental design consisted of FMTs twice weekly for 12 weeks. It might not represent an FMT transfer and microbiota colonisation experiment but rather a continuous exposure to human fecal samples, therefore, these evidences should be presented in the context of the original experiments. Alternatively, the authors might want to consider removing this section since it is redundant with findings by Pernigoni et al., that are described later by the authors.

Ans: Thank you for your valuable suggestions. We removed the section about the work by Yufei et al. from the revised manuscript.

Lines 310-311; This sentence is to general, please remove.

Ans: We agree with your suggestion, and removed the sentence in line 310-311.

Pernigoni et al., did not report reduced TRAMP-C1 tumor growth with prevotella stecorea this is must be corrected.

Ans: We corrected the sentences as followings.

FMT with hormone-sensitive microbiota or administration of P. stercorea can decrease androgens levels in CTX mice and delay the onset of CRPC.(Pernigoni)

The authors are describing chicken and egg type of data, with sex hormones driving changes in the microbiota and gut microbes differentially impacting testosterone levels. Please provide a comprehensive assessment of the dynamics between hormones and bacteria and possibly include a figure to help with the general understanding of it.

Ans: We appreciate your suggestion. We added the following figure which show the association between gut microbiome and ADT.

+Conclusion:

There are many issues with the conclusion that require significant rewording.

Why is “Conclusions” plural?

Ans. Thank you for the advice. We changed “Conclusions” to “Conclusion”.

The statement: “The gut microbiome is greatly influenced not only by our genetic makeup” is not factual. The current data rather support the opposite, namely that the greater proportion of gut microbial populational changes is influenced by the environment.

What do the authors mean by “lifestyle choices”?

Ans: We changed it according to the reviewer’s advice as following.

Conclusion (Line344-345)

The gut microbiome is greatly influenced by several environmental factors, such as lifestyles.

The authors state that: “Dysbiosis of the gut alters microbial composition which can in turn promote prostate carcinogenesis and progression.” I am not convinced that dysbiosis apply to microbiota changes reported in prostate cancer patients thus far and there is no convincing data as of yet, on prostate carcinogenesis with regard to a possible causal role of the gut microbes.

Ans: We appreciate your opinion. We changed the relevant text as follows.

Conclusion (Line345-346)

Change of the gut microbiota can be involved in prostate cancer progression through its metabolites and endotoxins.

Prebiotics and probiotics were never introduced or discussed in the main text, please describe in the text or remove from the conclusion.

Ans. We agree your opinion. The text about prebiotics and probiotics was removed.

Minor points:

Line 12 and 61, present tense for: “hypothesized”

Ans. We corrected it.

Line 39, typo: “Prostate cancer develops due the mutation”

Ans. We corrected it.

Line 118: consider changing “host location” to “individual’s geographic location”

Ans. We corrected it.

Line 135 and 138: weird characters

Ans. We corrected them.

Line 152, what are “local immune systems” referring to?

Ans. We revised it as following;

Gut microbes have direct contact with the intestinal wall, consequently, several intestinal diseases are affected by gut microbiota directly or indirectly through the modulation of local immune systems, such as regulatory T cell, dendric cells and CD4+ T cell[40].

Line 352: why thanking the laboratory and hospital staff for collecting, storing and managing samples?

Ans. We removed it.
